# Personal and Household Hygiene Measures for Preventing Upper Respiratory Tract Infections among Children: A Cross-Sectional Survey of Parental Knowledge, Attitudes, and Practices

**DOI:** 10.3390/ijerph20010229

**Published:** 2022-12-23

**Authors:** Man-Wai Leung, Margaret O’Donoghue, Lorna Kwai-Ping Suen

**Affiliations:** 1Faculty of Health and Social Sciences, The Hong Kong Polytechnic University, Hong Kong; 2School of Nursing, The Hong Kong Polytechnic University, Hong Kong; 3School of Nursing, Tung Wah College, Hong Kong

**Keywords:** hygiene, upper respiratory tract infection, parents, children, COVID-19

## Abstract

Personal and household hygiene measures are important for preventing upper respiratory tract infections (URTIs) and other infectious diseases, including coronavirus disease 2019 (COVID-19). An online survey recruited 414 eligible parents in Hong Kong to study their hygiene knowledge, attitudes, and practices (KAPs) regarding the prevention of URTIs among their children. The average knowledge score was high (10.2/12.0), but some misconceptions were identified. The majority of the participants agreed that good personal hygiene (93.5%) and good environmental hygiene (92.8%) can prevent URTIs. The average score for hand hygiene practices was high (3.78/4.00), but only 56.8% of the parents always performed hand hygiene before touching their mouths, noses, or eyes. In terms of environmental hygiene, only some household items were disinfected with disinfectants (door handles in 69.8% of the households, toilet seats in 60.4% of the households, the floor in 42.8% of the households, dining chairs in 24.2% of the households, and dining tables in 20.5% of the households). A higher knowledge score was associated with parents having tertiary educational levels or above, working as healthcare professionals, living in private residential flats or staff quarters, or having household incomes of HKD 70,000 or above. The results of multiple regression analyses also indicated that parents who were healthcare professionals and with higher household income had a better parental knowledge of hygiene measures after adjusting the attitude score. For hand hygiene, parents who achieved higher attitude scores obtained higher practice scores. Under the fifth wave of the COVID-19 epidemic, there were some misconceptions regarding hygiene among parents. Any health promotion program should target parents regarding taking proper personal and household hygienic measures, especially for those who had relatively lower socio-economic status and/or from a non-healthcare background. Motivating attitudes toward hand hygiene can lead to better practices.

## 1. Introduction

Upper respiratory tract infections (URTIs) are usually self-limited diseases that require only supportive management [1]. However, they are an important cause of childhood morbidity and a major cost to society [2,3,4,5]. In some cases, URTIs can lead to more serious complications, such as acute otitis media, asthma exacerbations, bronchiolitis, and pneumonia [6]. Most upper respiratory tract infections are caused by viruses, including rhinoviruses, which account for at least 30% of the cases in children under 5 years of age [7]. Other viral causes include the respiratory syncytial virus, the influenza virus, parainfluenza, human metapneumovirus, adenoviruses, and coronaviruses [7]. Most infants and young children have, on average, 6 to 10 such illnesses a year [8,9]. The modes of transmission for these viruses include droplets [10], direct contact with infectious secretions, and indirect transmission through contaminated hands [11].

Adherence to appropriate personal and household hygiene practices has been recommended as an important non-pharmaceutical intervention for preventing outbreaks of URTIs, such as pandemic influenza [12]. When there is an emerging respiratory infectious disease and effective drugs for treatment or vaccines for prevention are unavailable in the short term, preventive strategies would rely solely on non-pharmaceutical measures [13]. During the outbreak of severe acute respiratory syndrome (SARS) in 2003 and the human influenza (H1N1) pandemic in 2009, public health organizations emphasized good personal hygiene, as well as adequate environmental decontamination, including household cleansing and disinfection [14,15,16,17]. Such practices have demonstrated benefits in preventing upper respiratory tract infections and seasonal influenza [12,18,19]. The COVID-19 pandemic, which began in 2020, once again highlighted the importance of personal and home hygiene practices for preventing respiratory tract infections. Adherence to personal and household hygiene were recognized as two of the most important non-pharmaceutical interventions for preventing infection [20,21,22]. SARS-CoV-2 transmission was reported to be the highest in situations where there was sustained and prolonged contact among people, such as in the household setting, when compared with the travel, workplace, or casual close-contact environment [23]. However, studies revealed that adherence to hygiene practices can vary over time [24,25].

Children, particularly those less than 5 years old, suffer from upper respiratory tract infections more frequently than older children or adults, as they are still in the stage of learning hygiene practices [26,27]. Parents play an important role in protecting their children from harm, including from diseases such as respiratory infections. They can educate children on hygiene behaviors, such as hand hygiene, to avoid infectious diseases. A study showed that the hand-washing practices of parents are significantly correlated with the hand-washing behaviors of their children [28]. In addition to modeling good personal hygiene, parents can aim to provide a hygienic home environment for their children [29,30].

Studies demonstrated that appropriately using household disinfectants, regularly cleaning the home, and ensuring fresh air circulation are important hygiene measures to protect children from respiratory tract infections [11,31,32,33]. However, parents’ hygiene knowledge, attitudes, and practices (KAPs) may sometimes be limited due to poor education or other demographic factors and their adherence to best practices may vary over time. There appears to be a lack of literature on parental knowledge, attitudes, and practices related to hygiene for preventing URTIs, especially at the household level.

The recent COVID-19 pandemic is a reminder of the importance of awareness and education for the prevention of URTIs. This is an appropriate time to conduct a study to determine parental KAPs related to personal and home hygiene for preventing URTIs and to identify any knowledge gaps or misunderstandings that could lead to less-than-optimal home and/or personal hygiene outcomes. This study aimed to determine the parental KAPs related to hygiene (hand hygiene, environmental disinfection, and indoor ventilation) and how these variables are associated with the prevention of URTIs in their children and socio-demographic characteristics.

## 2. Materials and Methods

### 2.1. Study Design and Participants

A cross-sectional online survey was conducted in a community-based setting in Hong Kong.

The inclusion criteria for the study required that the participants (1) had one or more children aged 16 years or below, (2) were residing with their children, and (3) were the family member that was mainly responsible for maintaining family hygiene and a clean home environment.

According to the Census and Statistics Department, Hong Kong has 2.7 million domestic households with an average domestic household size of 2.7 members [34]. A sample size of 385 parents was calculated using the Cochrane formula for sample size, with a 95% confidence level and a 5% margin of error [35]. Convenience and snowball sampling methods were used to enroll potential participants, who were contacted through the personal and professional network of the researcher. The questionnaire link and the invitation to participate in the study were shared through email and WhatsApp, and respondents were encouraged to forward the link to their friends and colleagues.

### 2.2. Survey Instrument

An online structured questionnaire with closed-ended questions was developed following an extensive review of the literature [35,36,37,38,39,40,41,42,43]. The researcher’s clinical experience as a 15-year-specialty-experience infection control nurse working with parents of children hospitalized with respiratory tract infections was incorporated into the instrument. The questionnaire was bilingual (Chinese and English). Prior to commencing the study, two groups of subject experts were invited to check (a) the content validity and (b) the translation equivalence of the instrument. These expert panels included a professor from the academic field (nursing), one clinical microbiologist, a pediatrician, and two infection control nurses. Once the expert panel review was completed, the overall content validity index was calculated to be 96.0% and the translation equivalence was 97.0%. Minor adjustments were made to the questionnaire on the basis of the feedback from the subject experts. To determine the test–retest reliability, ten parents (nine mothers and one father) were invited to complete the questionnaire two times over two weeks. Intraclass correlation coefficients were satisfactory: section I (knowledge): 0.85; section II (attitudes): 0.72; and section III (practices): 0.81. However, in the case of one respondent, the results deviated when compared with those of other participants. Further investigation revealed that this respondent was not the family member that was mainly responsible for household hygiene. Thus, to ensure that only the appropriate subjects participated in the study, the inclusion criteria for the final version of the questionnaire were revised to clarify this and the following question was added to the online survey: Are you the family member who is mainly responsible for maintaining the hygiene of your family and the cleanliness of your home environment? Only those who responded with a ‘yes’ could proceed to complete the survey.

The final questionnaire was composed of four sections: 

I—Knowledge: A total of 12 questions (of which 10 were the true/false type and two were multiple choice questions with one correct answer) were used to measure parental knowledge of hygiene measures for preventing URTIs.

This section investigated parental knowledge about hygiene practices, including hand hygiene, household hygiene, and indoor ventilation, for preventing URTIs. Of the questions, 11 were constructed as positive statements and 1 was designed as a negative statement. Each correct answer was awarded 1 point, while wrong answers scored 0 points. The total obtainable maximum score was 12, while the minimum score was 0.

II—Attitudes: This section included five questions related to parental attitudes about personal and household hygiene for preventing URTIs in children. All questions were constructed as positive statements. A 5-point Likert scale (strongly disagree, disagree, neutral, agree, and strongly agree) was used. The maximum score obtainable was 25 and the minimum score obtainable was 5. Higher scores indicated more positive attitudes.

III—Practices: This section included five questions with 20 sub-questions. 

Three main aspects of hygiene practices were measured: (i) Hand hygiene (five questions investigating the parents’ hand hygiene practices and five questions exploring the parents’ practices for encouraging their children to perform hand hygiene; all questions were constructed as positive statements). A 4-point Likert scale was used for this section (never, rarely, sometimes, and always). (ii) Cleaning and disinfection of the home. This consisted of five questions on the type of cleaning performed, if any. Participants could specify whether they never cleaned the indicated fixtures in their homes; wipe them with water only; clean them with water and a common household cleaner, e.g., a detergent; or used a disinfectant to decontaminate them (if this option was selected, participants were required to answer an additional question on the type of disinfectant used and five questions on the frequency of cleaning or disinfecting the home, for which, participants could choose from a range of frequencies: once per day, 2 to 6 times per week, etc.). The total obtainable maximum score for hygiene practice was 83 and the minimum score was 17. (iii) Indoor ventilation: There was one question each on the frequency of opening windows in the home and the use of fans. A 4-point Likert scale was used, as above.

IV—Demographics: This section contained 14 questions and gathered information on the participants’ socioeconomic status, including gender, age, marital status, educational level, occupation, size of residence, and household income. Additional questions determined whether the respondent was the mother or father; the number of other family members in the home; the number of children; the age of the children; whether the children were in a day care center, in kindergarten, or in school; and whether the children had any chronic health conditions.

### 2.3. Data Collection

An electronic online survey was created using the Google online survey platform, which is a free online survey tool that maintains data privacy. No participant identifiers, such as name or Hong Kong identity card number, were collected. A brief description of the study, the information sheet, and the consent form were provided in the link. Before they completed the questionnaire, participants were requested to read the study information sheet and provide informed consent. 

The survey was accessible online for four weeks after the announcement. Two reminders were sent, one after week 1 and one after week 2. The study was conducted from 26 March to 21 April 2022. 

### 2.4. Statistical Analysis

IBM SPSS Statistics for Windows, version 29.0, was used for the data analysis. Categorical variables in demographics were analyzed using descriptive statistics, while continuous variables, such as knowledge scores, were reported as the mean +/− the standard deviation (SD). Continuous variables, such as knowledge scores and demographic characteristics, were examined using the Mann–Whitney U and Kruskal–Wallis H, while chi-square was used to analyze the attitude and practice scores. Associations between KAPs were tested with Spearman’s rank correlation. Stepwise multiple regression was conducted to identify the most parsimonious combinations of attitude and other extraneous variables regarding predicting the knowledge score. The statistical significance level was set to *p* < 0.05, and all statistical tests were two-tailed.

### 2.5. Ethical Considerations

Ethical approval for the study was obtained from the Human Subjects Ethics Sub-committee of the Hong Kong Polytechnic University (reference number: HSEARS20220210001). Bilingual information sheets and consent forms (in English and Chinese) were provided online. Participation was on a voluntary basis and the questionnaires were anonymous. Respondents could decline to answer any question and could opt to withdraw from the study at any point.

## 3. Results

### 3.1. Participant Characteristics

A total of 507 parents consented to participate in the survey. Of these, 414 (82%) satisfied the eligibility criteria.

The majority were mothers (84%) and were aged between 35 and 44 years (63%). Most (72%) had received tertiary education or above. Almost all respondents were married (95%) and 36% were full-time homemakers. 

Most households (87%) had between three and five members, and more than half (53%) had two children. The majority of respondents lived in private dwellings (68%), while 15% resided in public housing. The average monthly household income was HKD 50,000 or above for 53% of the households. Table 1 presents further details of the participants’ demographics.

### 3.2. Knowledge of Hygiene Measures

The average knowledge score obtained was 10.2 (SD = 1.4, range 5 to 12). The range of correct answers for the individual knowledge items was between 60.1% and 99.8%. Table 2 presents the results of percentage of correct and incorrect answers.

Almost all participants were aware that touching one’s face, eyes, mouth, or nose with unwashed hands could increase the risk of contracting a URTI (99.8%) and that keeping windows open could help in maintaining good indoor ventilation in the home (99.8%). However, some respondents were confused about some aspects of hand hygiene: of the respondents, 18% thought that alcohol-based hand rubs (ABHRs) were more effective than hand washing, 24% were unaware that ABHRs may be ineffective when the hands are visibly soiled, 12% did not know the recommended concentration of alcohol in ABHRs, and 14% did not know the duration of hand washing required to achieve effective hand hygiene.

Almost 40% of the participants thought that disinfectants were more effective in preventing URTIs if they were sprayed on environmental surfaces rather than applied with a cloth.

Differences in total knowledge scores among different demographic characteristics were analyzed using the Mann–Whitney U and Kruskal–Wallis H. Table 3 presents the results. There were significant differences in knowledge scores related to parental levels of education, occupation, types of housing, and average monthly household incomes. Higher knowledge scores were obtained by parents who were healthcare professionals, had a household income of HKD 70,000 or above, and had achieved a tertiary level of education. Lower knowledge scores were obtained by older parents (55 to 64 years old). 

### 3.3. Parental Attitudes to Hygiene for Preventing URTIs

Five questions measured parental attitudes toward hygiene for preventing URTIs in children. The results are presented in Table 4. The average attitudes score was 4.32 (SD = 0.81) out of a maximum of 5, indicating that participants had a relatively positive attitude toward personal and household hygiene as measures to prevent URTIs in children. Participants rated their own hygiene habits and their supervision of their children’s hygiene habits positively (88% and 91%, respectively). The majority of participants believed that good personal hygiene and good environmental hygiene in the home could prevent URTIs (93.5% and 92.8%, respectively).

Associations between parental attitudes and demographic characteristics were analyzed using chi-square. Association between parental attitudes and children with health problems was statistically significant (*p* < 0.05). The association between attitudes and average monthly income (<HKD 20,000 compared with ≥HKD 20,000) was statistically significant (*p* < 0.05). 

### 3.4. Parental Practices Related to Hygiene Measures in URTI Prevention

This section presents the results of participants’ practices for hand hygiene, household hygiene, and maintaining indoor ventilation.

The mean score for hand hygiene practices was 3.78 (SD = 0.31) out of a maximum of 4, indicating that participants regularly implemented the correct practices for performing hand hygiene and encouraged their children to do the same. Almost all (96%) participants claimed to always perform hand hygiene after using the toilet, and 92% always encouraged their children to do the same. Only 57% of the participants always performed hand hygiene before touching their mouths, noses, or eyes, and 72% always encouraged their children to perform hand hygiene before touching those body parts. The average score for cleaning and disinfecting the home (using a disinfectant) was 3.28 (SD = 0.49) out of a maximum of 4, indicating that the home was carefully cleaned with water and common household cleaners (e.g., detergent). However, only 70% of the respondents used disinfectants to clean toilet seats and only 60% disinfected door handles. Only a small proportion used disinfectants to disinfect dining chairs (24.2%) and dining tables (20.5%). Bleach is a common household disinfectant. In this study, of the parents who used a disinfectant, 44% selected bleach to clean the toilet seat, 36% used bleach on door handles, 23% used bleach on the floor, 13% used bleach on dining chairs, and 8% used bleach on the dining table. The average score for cleaning and disinfecting the home (cleaning frequency) was 5.30 (SD = 0.96) out of a maximum of 7, indicating that most of the environment was cleaned/disinfected two to six times per week. For example, the percentages of participants who cleaned/disinfected various items at least once per week or more were as follows: door handles (77.8%), toilet seats (95.7%), floor (95.5%), dining chairs (81.4%), and dining tables (96.8%). The average score for indoor household ventilation was 3.51 (SD = 0.44) out of a maximum of 4, meaning that the participants generally maintained good indoor ventilation in houses. Most participants (79.2%) kept windows open at home. Detailed results of hygiene practices are presented in Table 5. 

### 3.5. Correlations between Parental Hygiene Knowledge, Attitudes, and Practices

The correlations between parental knowledge, attitudes, and practices related to household and personal hygiene for preventing URTIs among children were analyzed using Spearman’s rank correlation. There was no significant correlation between hygiene knowledge and attitudes. However, there was a significant correlation between hygiene attitudes and practices (*p* < 0.01).

### 3.6. Multiple Regression Analyses

Stepwise multiple regression (with probability of F-to-enter ≤0.50, probability of F-to-remove ≥0.100) was conducted to identify the most parsimonious combination of respondent (mother vs. father), age group (less than 18 years to 34 years as referent versus 35 to 44 years and 45 years or above), marital status (single/divorced/separated/widowed as referent versus married), education level (primary/secondary as referent versus tertiary or above), occupation (non-healthcare professional as referent versus healthcare professional and unemployed/retired/homemaker), average monthly household income (less than HKD 20,000 as referent versus HKD 20,000 to 49,999 and HKD 50,000 or above), attitude scores, and practice scores in predicting the parental knowledge of hygiene measures for preventing URTIs. After conducting the stepwise regression, the model that included healthcare professional as the occupation, average monthly household income of more than HKD 50,000, and attitude scores toward hygiene measures was found to be the most parsimonious (F(3, 410) = 7.486, *p* < 0.001). The final model presented in Table 6 suggests that respondents who were healthcare professionals had an increase in knowledge score regarding hand hygiene of 0.532 relative to those who did not have a healthcare background, and those with an average monthly household income of HKD 50,000 or above had an increase in knowledge score of 0.452 relative to those in the referent group (i.e., average household income less than HKD 20,000) after adjusting for the attitude score. The correlation coefficient between attitude and practice was highly statistically significant (r = 0.276, *p* < 0.001), and the final model using stepwise regression analyses to evaluate the predictor(s) for practice score indicated an increase in practice score of 0.292 per each attitude score increase (F(1, 412) = 29.091, *p* < 0.001; Table 7). The assumptions of linearity, data multicollinearity, homoscedasticity and distribution of residuals, and absence of influential cases were checked and met.

## 4. Discussion

To the best of our knowledge, this is the first study that investigated parental knowledge, attitudes, and practices related to household and personal hygiene for preventing URTIs in children. The majority of parents in this study appeared to be knowledgeable about the benefits of hand and home hygiene in preventing URTIs in their children. 

In this study, the hygiene knowledge scores of parents were relatively high. Parents with tertiary or higher educational levels, working as healthcare professionals, living in private flats or staff quarters, or with high average monthly household incomes obtained higher hygiene knowledge scores. According to the most recent census and statistics report, the average monthly household income in Hong Kong is HKD 28,000 [34]. In our study, parents with higher educational levels and above-average incomes had better knowledge scores. Similar findings were reported for the general community in Hong Kong following the SARS outbreak in 2003. In a study conducted by the Department of Health to determine knowledge, attitudes, and practices of the general public related to personal, food, and environmental hygiene, respondents who had higher education levels and higher household incomes also demonstrated better hygiene knowledge [44]. This suggests that educational material promoting hygiene knowledge may be presented in a manner that is too complicated or there may be too much information provided; because of this, those who are at lower income levels or not well educated may not be able to understand or memorize the key elements. Therefore, education regarding hygiene knowledge should be tailored for various social groups and key aspects of hygiene knowledge on URTI prevention should be presented in an understandable format for all population groups. In addition, findings arising from multiple regression analysis indicated that hygiene knowledge scores of parents working as healthcare professionals were significantly higher compared with those of parents in other occupations. In another KAP study on COVID-19 information and prevention, healthcare-related occupation was significantly associated with a higher knowledge score [45]. Hygiene (such as hand hygiene and environmental disinfection) as one of the important preventive measures of infectious diseases is included in the training of healthcare professionals; parents working in healthcare fields should have better hygiene knowledge than other parents. Parents living in private residential flats or staff quarters obtained higher hygiene scores, which may be related to them having a higher household income than others. In our study, 85% (240/281) of the parents living in private flats and 74% (14/19) of the parents living in staff quarters had monthly incomes of more than HKD 30,000. In comparison, only 40% (25/63) of those living in public housing had such high incomes. Since parents in public housing obtained lower hygiene knowledge scores, more health education programs should be conducted in such areas.

Although hand hygiene knowledge was good, some misconceptions were identified with regard to the length of time required for washing hands in order to achieve adequate hand hygiene. This confusion could result in incorrect or incomplete messaging by the parents to their family members, including children. There have been ongoing campaigns, both locally and worldwide, for more than 20 years to improve hand hygiene [46]. However, misconceptions still arise. For example, in our study, some parents thought that alcohol-based hand sanitizers are effective at killing germs even when the hands are visibly dirty or greasy and some parents did not know that the recommended percentage of alcohol in sanitizers is 70% to 80%. In hand hygiene promotion activities, correct and key messages should be emphasized. Misconceptions revealed about the correct use of disinfectants (such as the benefits of disinfectant sprays over liquid applied to a surface) highlight the risk that the home may not be appropriately disinfected. Clear instructions on the correct use of disinfectants in the home is critical for preventing respiratory diseases, especially during epidemics [47]. 

In this study, overall, high scores were achieved for attitudes toward hygiene measures. The parents believed that good personal and environmental hygiene could prevent URTIs. One of the reasons for the high scores in attitudes toward hygiene measures may have been the impact of COVID-19 locally in Hong Kong, as the levels of anxiety regarding an infectious disease, especially an outbreak, could influence personal hygiene attitudes [48]. A local study found that there was a sharp increase in fear in early February 2022 and that anxiety escalated in the early stages of the rapid spread of the COVID-19 epidemic [49]. As the HKSAR government and other media significantly promoted the importance of hygiene measures for COVID-19 prevention, parents had a strong positive attitude toward hygiene measures in this special situation of the COVID-19 outbreak.

Parental attitudes toward hygiene measures were associated with children with health problems. Parents with children who have chronic diseases are likely to experience more fear and anxiety, and their attitudes toward protecting children might be affected in the current COVID-19 pandemic [50]. Healthcare workers and healthcare institutions may need to address the special health needs or educational demands of parents with children who have chronic diseases. 

Both parents and their children practiced good hand hygiene to prevent URTIs. Among the moments of hand hygiene, the highest scores were obtained for performing hand hygiene after using the toilet (parents: 3.95, SD = 0.290; children: 3.91, SD = 0.307). This finding is consistent with the local survey conducted following the SARS outbreak by the Department of Health to determine knowledge, attitudes, and practices of the general public related to personal, food, and environmental hygiene [44]. However, although 99.8% of the participants knew that touching their faces, eyes, mouths, or noses with contaminated hands could increase their risk of contracting URTIs, performing hand hygiene before touching the mouth, nose, or eyes received the lowest score among parents (3.51, SD = 0.610) and children (3.68, SD = 0.533). Only 56.8% of the parents always performed hand hygiene before touching their mouths, noses, or eyes, while only 71.7% of the parents always encouraged their children to perform hand hygiene at this high-risk moment. A systematic review conducted to determine the frequency at which humans touch the T-zone (eyes, nose, mouth, and chin) reported that touching the face is a type of consistent regulatory movement across the world and high community awareness and extensive behavioral interventions must be ensured to control this movement, especially during a pandemic of infectious disease [51]. Drawback challenge to non-pharmaceutical interventions is the fact that people with hygiene knowledge may not necessarily follow hygiene practices.

Various studies showed that toilet seats [52]; door handles [52,53,54,55,56]; floors [55]; and furniture, such as tables and chairs [52,54,55], in household and other settings could be contaminated with respiratory viruses. The importance of environmental cleaning and disinfection was highlighted. In our study, the observed practice score suggested that the homes were well cleaned but imperfectly disinfected with appropriate disinfectants, such as diluted household bleach. In the 2005 local survey, 40.9% of respondents used a 1:99 dilution of household bleach when they disinfected their homes [44]. Our study results were similar and we also observed that parents used disinfectants more frequently for some household items than others. The toilet was perceived as a dirty area and was disinfected with bleach more frequently than the dining table or chairs, which may be considered clean in comparison. In a pandemic situation, all frequently touched household surfaces should be disinfected rather than just cleaned with detergents, especially if there is an infected person in the household. Those items perceived as clean might still be contaminated with pathogens. The correct and frequent use of disinfectants should be highlighted in infectious disease prevention programs.

For the cleaning frequency of the household environment, in each home, some environmental surfaces were relatively frequently cleaned. It should be noted that most people in Hong Kong live in apartments with an average size of 45 square meters. Therefore, cleaning a home in Hong Kong is not time consuming and each family cleans the home daily or weekly. In our study, only 34% of the parents reported that the door handles in their houses were cleaned once per day or more, even in the fifth wave of the COVID-19 pandemic. However, the cleaning frequency was higher than that in the non-pandemic time. In the 2005 survey, only around half of the people in Hong Kong cleaned their homes three or more times per week, 22.8% cleaned their homes twice weekly, and 21.1% cleaned their homes once per week [45]. In our study, parents said that many household surfaces were cleaned at least once per day (58.5%, 59%, and 78% of the households cleaned toilet seats, floor, and dining tables, respectively, at least once per day). Although environmental cleaning can be considered a tedious task, the outbreak situation may have increased the cleaning frequency of the household environment and surfaces.

In our study, the parents generally maintained good indoor ventilation in their homes. Overall, 79.2% of the parents kept the home windows open. This is fully consistent with the 2005 survey in which 73.1% of the respondents always kept windows at home open to maintain good indoor ventilation [44]. As keeping windows open does not cost anything, on finding out that this practice is necessary, most people in Hong Kong comply with this hygiene measure.

For personal and household hygiene practices, there were no significant statistical differences by housing type and size. Our study did not find any association between housing demographics and the hygiene practices of the households in Hong Kong. This means that irrespective of house size, people may still be able to perform good personal and environmental home hygiene, but this aspect should be explored further in future studies.

In our study, parental hygiene knowledge was not significantly associated with attitudes or practices. One of the reasons could have been that our survey questions on hygiene knowledge were relatively challenging and many parents may have found it difficult to answer the questions (only 75 out of the 414 participants, or 18%, achieved full marks in their answers to the 12 questions). Even with inadequate hygiene knowledge, parents could still have positive hygiene attitudes and good hygiene practices. In addition, good knowledge might not directly translate into good hygiene behaviors because other factors, such as a busy lifestyle and well-developed healthcare facilities, may hinder compliance to preventive behaviors [57].

Our study found that the parents’ attitude toward hygiene could affect hygiene practices. Regression analyses also found that an increase in parental attitude scores toward hand hygiene could lead to an increase in practice scores. These findings are similar to those of another cross-sectional study targeting parents of preschool children in Malaysia [58]. Our findings are also consistent with positive correlations between attitudes toward health and disease-preventive measures in other current studies, but these studies did not focus on parents [48,59].

Our study had several limitations. (1) Snowball and convenience sampling were used and the study was limited to parents in Hong Kong. Thus, we may not be able to generalize the findings to parents in other cities or countries. In addition, only around 10% of the parents in the sample had average household incomes below HKD 20,000. Thus, the study might not reflect the picture of parents in Hong Kong with low incomes. (2) Our sample size was not too large. However, our findings could serve as a baseline parameter for larger-scale studies in the future. (3) The online questionnaires were self-reported and hygiene practices were not validated. (4) Hong Kong faced the fifth wave peak from February to March of 2022 and our parental KAP findings may have been affected by this special situation. Had this not been a rapid study, a longitudinal study approach may have overcome these problems. (5) For future research, variables such as the state of health of participants and children could be included in the survey so that any risk association between those who become ill more often and those who have less or greater hygiene knowledge or practice hygiene less or more could be explored.

## 5. Conclusions

Parents can carry out simple and basic personal and household hygiene measures to reduce their children’s risk of contracting URTIs and other respiratory diseases. The parents in Hong Kong who participated in our study had generally good hygiene knowledge about URTI prevention but still had certain misconceptions. The majority of the parents had a strong positive attitude toward hygiene measures. Parents and their children practiced good hand hygiene, but some did not comply with hand hygiene moments, such as before touching their mouths, noses, and eyes. Disinfectants were used on environmental surfaces of the home, but the frequency of this practice depended on the specific household items or surfaces. Disinfectants were used less often on some items, such as dining tables and chairs, which were deemed by participants to be clean. Some parents did not clean their household items frequently, even during the fifth wave of the COVID-19 epidemic in Hong Kong. Any health promotion program should encourage parents to take proper personal and household hygiene measures and should ensure to provide information that can be understood by those of a relatively lower socio-economic status, and/or those from non-healthcare background. Specific targeted programs could be considered for parents of children with chronic illnesses. Attitudes motivation is important to lead to better hygiene practices.

## Figures and Tables

**Table 1 ijerph-20-00229-t001:** Demographics of the participants.

Demographics	Characteristics	Number	(%)
Category	Mother	348	84
	Father	66	16
Age	Less than 18 years	3	0.7
	18 to 24 years	1	0.2
	25 to 34 years	49	11.8
	35 to 44 years	259	62.6
	45 to 54 years	98	23.7
	55 to 64 years	3	0.7
	65 years or above	1	0.2
Marital status	Single	2	0.5
	Married	392	94.5
	Divorced/separated/widowed	20	5.0
Level of education	Primary	1	0.2
	Secondary	114	27.5
	Tertiary or above	299	72.2
Occupation	Full-time homemakers	149	36
	Retired persons	3	1
	Office/clerical work	47	11
	Professionals (non-healthcare related)	58	14
	Professionals (healthcare related)	49	12
	Managers/administrators	59	14
	Others	48	12
Number of people living in the same household (including foreign domestic helpers)	2	14	3
	3	88	21
	4	160	39
	5	114	28
	6	30	7
	7 or above	8	2
Number of children	1	161	39
	2	220	53
	3	32	8
Type of housing	Public housing	63	15
	Housing Authority/Society subsided sale flat	35	8
	Private residential flat	281	68
	Village house	19	5
	Staff quarter	11	3
	Others	4	1
Size of house	Less than 20 square meters	9	2
	20 to 29.9 square meters	30	7
	30 to 39.9 square meters	78	19
	40 to 49.9 square meters	101	24
	50 to 59.9 square meters	85	21
	60 to 69.9 square meters	43	10
	70 square meters or above	68	16
Average monthly household income	HKD 5000 to 9999	14	3
	HKD 10,000 to 14,999	7	2
	HKD 15,000 to 19,999	21	5
	HKD 20,000 to 24,999	25	6
	HKD 25,000 to 29,999	23	6
	HKD 30,000 to 49,999	99	24
	HKD 50,000 to 69,999	84	20
	HKD 70,000 or above	136	33

**Table 2 ijerph-20-00229-t002:** Parental knowledge of the importance of personal and home hygiene in preventing URTIs in children.

Question				True N (%)		False N (%)
1. Using an alcohol-based hand sanitizer is better than washing hands with soap and water in preventing upper respiratory tract infection.				74 (17.9%)		340 (82.1%) *
2. An alcohol-based hand sanitizer is still effective in killing germs when hands are visibly dirty or greasy.				101 (24.4%)		313 (75.6%) *
3. Washing hands with cold water is as effective as using hot water for cleaning hands.				314 (75.8%) *		100 (24.2%)
4. Touching one’s face, eyes, mouth, or nose with unwashed hands can increase the risk of one getting upper respiratory tract infection.				413 (99.8%) *		1 (0.2%)
5. Alcohol can be used for disinfecting metallic surfaces.				295 (71.3%) *		119 (28.7%)
6. Disinfectants are more effective in preventing upper respiratory tract infection when they are sprayed on environmental surfaces rather than applied with a cloth.				165 (39.9%)		249 (60.1%) *
7. All cleaning products are also disinfectants.				41 (9.9%)		373 (90.1%) *
8. Household bleach can be used to disinfect the home to protect against the germs causing upper respiratory tract infections.				381 (92.0%) *		33 (8.0%)
9. Keeping windows open can help in maintaining good indoor ventilation at home.				413 (99.8%) *		1 (0.2%)
10. If an air purifier is used, no regular environmental disinfection is needed at home.				6 (1.4%)		408 (98.6%) *
Question	Answer N (%)					
11. The recommended percentage of alcohol in an alcohol-based hand rub is:	40–50%	70–80%	More than 90%, because a higher percentage of alcohol can kill germs better		Percentage does not matter in killing germs	
	24 (5.8%)	364 (87.9%) *	22 (5.3%)		4 (1.0%)	
12. What is the recommended duration for effective hand washing?	5 s	10 s	20 s		1 min	
	8 (1.9%)	32 (7.7%)	357 (86.2%) *		17 (4.1%)	

* Correct answer.

**Table 3 ijerph-20-00229-t003:** Association of knowledge score and demographic characteristics.

	Demographic Characteristics	Knowledge Score (SD)	Z/H	*p*-Value
Category	Mother	10.19 (1.43)	−0.61	0.540
	Father	10.21 (1.51)		
Age	25 to 34 years	10.29 (1.46)	5.92	0.115
	35 to 44 years	10.27 (1.38)		
	45 to 54 years	10.06 (1.53)		
	55 to 64 years	8.00 (2.00)		
Marital status	Single	10.50 (2.12)	2.16	0.340
	Married	10.22 (1.43)		
	Divorced/separated/widowed	9.70 (1.63)		
Level of education	Secondary	9.82 (1.54)	−3.08	0.002 **
	Tertiary or above	10.33 (1.39)		
Occupation	Retired persons	8.00 (2.00)	16.21	0.006 **
	Office/clerical work	10.23 (1.29)		
	Professionals (non-healthcare related)	10.26 (1.53)		
	Professionals (healthcare related)	10.78 (1.39)		
	Managers/administrators	10.05 (1.46)		
	Full-time homemakers	10.21 (1.34)		
	Others	9.79 (1.57)		
Type of housing	Public housing	9.73 (1.36)	13.98	0.016 *
	Housing Authority/Society subsided sale flat	10.11 (1.47)		
	Private residential flat	10.32 (1.42)		
	Village house	10.05 (1.55)		
	Staff quarter	10.36 (1.69)		
Size of house	Less than 20 square meters	9.22 (1.92)	10.18	0.117
	20 to 29.9 square meters	9.87 (1.55)		
	30 to 39.9 square meters	10.06 (1.31)		
	40 to 49.9 square meters	10.15 (1.42)		
	50 to 59.9 square meters	10.26 (1.59)		
	60 to 69.9 square meters	10.35 (1.31)		
	70 square meters or above	10.50 (1.36)		
Average monthly household income	Less than HKD 5000	9.80 (1.48)	21.21	0.007 **
	HKD 5000 to 9999	9.50 (2.18)		
	HKD 10,000 to 14,999	10.14 (1.22)		
	HKD 15,000 to 19,999	9.71 (1.77)		
	HKD 20,000 to 24,999	9.88 (1.24)		
	HKD 25,000 to 29,999	9.48 (1.70)		
	HKD 30,000 to 49,999	10.12 (1.29)		
	HKD 50,000 to 69,999	10.26 (1.38)		
	HKD 70,000 or above	10.54 (1.37)		

* *p* < 0.05, ** *p* < 0.01.

**Table 4 ijerph-20-00229-t004:** Parental attitudes on hygiene measures for URTI prevention.

Question	Strongly Disagree N (%)	Disagree N (%)	Neutral N (%)	Agree N (%)	Strongly Agree N (%)	Mean (SD)
Good personal hygiene can prevent URTI.	17 (4.1%)	0 (0%)	9 (2.2%)	118 (28.4%)	270 (65.1%)	4.51 (0.89)
I believe I have good hand hygiene habits.	16 (3.9%)	0 (0%)	33 (8.0%)	218 (52.5%)	147 (35.4%)	4.16 (0.88)
I try to ensure my children have good hand hygiene habits.	15 (3.6%)	0 (0%)	22 (5.3%)	233 (56.3%)	144 (34.8%)	4.19 (0.83)
Good environmental hygiene at home can prevent URTI.	16 (3.9%)	3 (0.7%)	10 (2.4%)	154 (37.1%)	231 (55.7%)	4.40 (0.89)
I feel more comfortable when my home is clean and free of germs.	17 (4.1%)	3 (0.7%)	18 (4.3%)	159 (38.3%)	217 (52.3%)	4.34 (0.92)

**Table 5 ijerph-20-00229-t005:** Parental hygiene practices for URTI prevention in their children.

**Questions**		**Never** ** N (%)**	**Rarely** ** N (%)**	**Sometimes** ** N (%)**		**Always** ** N (%)**		**Mean (SD)**
I perform hand hygiene								
Before touching my mouth, nose, or eyes		2 (0.5%)	19 (4.6%)	158 (38.2%)		235 (56.8%)		3.51 (0.61)
Before eating		1 (0.2%)	8 (1.9%)	56 (13.5%)		349 (84.3%)		3.82 (0.45)
After using the toilet		2 (0.5%)	1 (0.2%)	14 (3.4%)		397 (95.9%)		3.95 (0.29)
When my hands are contaminated by respiratory secretions after coughing or sneezing		1 (0.2%)	4 (1.0%)	68 (16.4%)		341 (82.4%)		3.81 (0.43)
After disposing of soiled tissues after blowing my nose		1 (0.2%)	7 (1.7%)	94 (22.7%)		312 (75.4%)		3.73 (0.50)
I encourage my children to perform hand hygiene								
Before they touch their mouths, noses, or eyes		0 (0%)	14 (3.4%)	103 (24.9%)		297 (71.7%)		3.68 (0.53)
Before they eat		1 (0.2%)	3 (0.7%)	53 (12.8%)		357 (86.2%)		3.85 (0.40)
After they use the toilet		1 (0.2%)	0 (0%)	33 (8.0%)		380 (91.8%)		3.91 (0.31)
When their hands are contaminated by respiratory secretions after coughing or sneezing		1 (0.2%)	6 (1.4%)	77 (18.6%)		330 (79.7%)		3.78 (0.47)
After they dispose of soiled tissues after blowing their noses		2 (0.5%)	8 (1.9%)	92 (22.2%)		312 (75.4%)		3.72 (0.52)
**Questions**		**I never clean them** ** N (%)**	**I wipe them with water only** ** N (%)**	**I clean them with water and a common household cleaner, e.g., a detergent** ** N (%)**		**I use a disinfectant to disinfect them** ** N (%)**		**Mean (SD)**
Door handles		18 (4.3%)	33 (8.0%)	74 (17.9%)		289 (69.8%)		3.53 (0.82)
Toilet seats		1 (0.2%)	12 (2.9%)	151 (36.5%)		250 (60.4%)		3.57 (0.56)
Floor		1 (0.2%)	27 (6.5%)	209 (50.5%)		177 (42.8%)		3.36 (0.61)
Dining chairs		20 (4.8%)	88 (21.3%)	206 (49.8%)		100 (24.2%)		2.93 (0.80)
Dining tables		0 (0%)	91 (22.0%)	238 (57.5%)		85 (20.5%)		2.99 (0.65)
**Questions**	**Never** ** N (%)**	**Less than once per month** ** N (%)**	**1 to 3 times per month** ** N (%)**	**Once per week** ** N (%)**	**2 to 6 times per week** ** N (%)**	**Once per day** ** N (%)**	**More than once per day** ** N (%)**	**Mean score (SD)**
Door handles: frequency of cleaning and disinfecting	16 (3.9%)	32 (7.7%)	44 (10.6%)	93 (22.5%)	89 (21.5%)	106 (25.6%)	34 (8.2%)	4.60 (1.55)
Toilet seats: frequency of cleaning and disinfecting	1 (0.2%)	2 (0.5%)	15 (3.6%)	66 (15.9%)	88 (21.3%)	183 (44.2%)	59 (14.3%)	5.47 (1.09)
Floor: frequency of cleaning and disinfecting	0 (0%)	7 (1.7%)	12 (2.9%)	58 (14.0%)	93 (22.5%)	182 (44.0%)	62 (15.0%)	5.49 (1.10)
Dining chairs: frequency of cleaning and disinfecting	17 (4.1%)	23 (5.6%)	37 (8.9%)	71 (17.1%)	105 (25.4%)	120 (29.0%)	41 (9.9%)	4.81 (1.53)
Dining tables: frequency of cleaning and disinfecting	1 (0.2%)	1 (0.2%)	11 (2.7%)	25 (6.0%)	53 (12.8%)	118 (28.5%)	205 (49.5%)	6.14 (1.09)
**Questions**	**Never** ** N (%)**	**Rarely** ** N (%)**		**Sometimes** ** N (%)**		**Always** ** N (%)**		**Mean (SD)**
I keep windows open at home	0 (0%)	13 (3.1%)		73 (17.6%)		328 (79.2%)		3.76 (0.50)
I switch on fans to enhance air flow at home	8 (2.0%)	35 (8.6%)		20 3(50.1%)		159 (39.3%)		3.27 (0.70)

**Table 6 ijerph-20-00229-t006:** The final model of regression analyses.

	Variables Entered	Unstandardized Coefficients		t	Sig.	Adjusted R^2^	F-Value (df) (Sig.)
		Beta	Std Error				
**Model**	(Constant)	9.167	0.386	23.764	<−0.001	0.047	F(3, 410)
	Occupation: healthcare professional #	0.560	0.219	2.556	0.011		=7.840,
	Monthly household income	0.448	0.141	3.164	0.002		*p* < 0.001 ***
	HKD 50,000 or above #						
	Attitudes toward hygiene	0.033	0.017	1.943	0.053		

# Dependent variable: total correct answers in knowledge questionnaire regarding hand hygiene. Occupation: healthcare professional (using non-healthcare occupation as the referent). Monthly household income: HKD 50,000 or above (using below HKD 20,000 as referent). *** *p* < 0.001.

**Table 7 ijerph-20-00229-t007:** The final model for predicting practice score regarding hand hygiene.

	Variables Entered	Unstandardized Coefficients		t	Sig.	Adjusted R^2^	F-Value (df) (Sig.)
		Beta	Std Error				
**Model**	(Constant)	54.873	1.187	46.240	<0.001	0.065	F(1, 412)
	Attitudes toward hand hygiene	0.292	0.054	5.394	<0.001		=29.091,
							*p* < 0.001 ***

Dependent variable: total practice score toward hand hygiene. *** *p* < 0.001.

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
