# Peer review of "Personal and Household Hygiene Measures for Preventing Upper Respiratory Tract Infections among Children: A Cross-Sectional Survey of Parental Knowledge, Attitudes, and Practices"

_ijerph, 2022, doi:10.3390/ijerph20010229_

Round 1

Reviewer 1 Report

Improvements in the clarity of methodology, data presentation, and discussion are suggested.

Author Response

Point 1:  Extensive editing of English language and style required. Improvements in the clarity of methodology, data presentation, and discussion are suggested. 

Response 1: Sincerely thank for your great comments and suggestions.  To improve the language and style, I've used the MDPI English Editing service. 

12/12: I believe the article now reads better following this editing exercise.

Reviewer 2 Report

The main question that the researchers express in their manuscript is to measure the knowledge of the main hygiene for the prevention of respiratory infections

The topic is not something new, however, it is necessary to have precise data on the weaknesses in the knowledge of the main biosafety measures to control the transmission of this type of respiratory virus, such as possible gaps identified, it could be important to include information in the survey on the state of health or on the frequency of respiratory diseases in order to be able to make a risk association between those who became more ill and those who had less or greater knowledge of the subject.

Incorporate data or refer to studies of the places where respiratory pathogens are mostly found in the environment, for example, door handles, hands, windows, and cell phone surfaces, among others.

If information is available, the points described above could be incorporated, in addition, the description of the methodology could be strengthened, including measures of central tendency, such as the median.

The conclusions are consistent with the evidence and arguments presented, although they should be strengthened in the discussion

The references are appropriate.

One or 2 graphs could be included that allow the results to be more visual.

Author Response

Point 1: Extensive editing of English language and style required.

Response 1: Sincerely thank for your tailored and good comments. To improve the language and style, I’ve applied for the MDPI English Editing service.   After editing, I found the article content is much better and I believed that this article is at an acceptable standard for publication.

Point 2: The main question that the researchers express in their manuscript is to measure the knowledge of the main hygiene for the prevention of respiratory infections. The topic is not something new, however, it is necessary to have precise data on the weaknesses in the knowledge of the main biosafety measures to control the transmission of this type of respiratory virus, such as possible gaps identified, it could be important to include information in the survey on the state of health or on the frequency of respiratory diseases in order to be able to make a risk association between those who became more ill and those who had less or greater knowledge of the subject.  Incorporate data or refer to studies of the places where respiratory pathogens are mostly found in the environment, for example, door handles, hands, windows, and cell phone surfaces, among others. 

If information is available, the points described above could be incorporated, in addition, the description of the methodology could be strengthened, including measures of central tendency, such as the median. 

Response 2: Your points are well taken.  We were unable to collect data on the state of health of participants such as frequency of respiratory diseases among parents and children for this study due to time constraints.  I have added your comments in our study's limitation and recommend future studies to include these important variables.

Point 3: One or 2 graphs could be included that allow the results to be more visual.  

Response 3: Apologies that we would like to keep the results presentation at this moment because the important results have been shown in tables and no specific data results are suitable to be presented as graphs. 

Reviewer 3 Report

This is an exciting manuscript that addresses an important area of public health; however, there are a few comments.

Introduction

Most upper respiratory tract infections are caused by viruses, including rhinoviruses which account for at least 30% of cases. This is an interesting figure; however, it will be important to categorically state if this substantial proportion is children or across all ages.

For a smooth and proper introduction flow, the transmission of the virus was highlighted in lines 40-42. Similarly, the transmission was further alluded to in Lines 56-58. I suggest you develop statements on the same topic in the same paragraph.

Materials and Methods

The sample size of 385 calculated in the study was based on household figures. Clarify if the 385 sample size refers to households or the number of parents or even children.

Section IV of the survey instrument in Lines 151 to 156 is supposed to be section I as it is the items that the participant will answer first.

The knowledge section had 12 items, and 10 were true or false. It will be essential if the response options of the 2 other items in the knowledge section are stated or based on what criteria they are answered.

For the attitude and practice sections, the score for each correct response needs to be stated.

State the total scores obtainable (minimum and maximum) for each of the knowledge, attitude and practice sections. 

Similarly, for all the sections, how many questions are constructed as positive statements and how many are negative statements (reverse statements)?

IBM SPSS Statistics for Windows, version 26 was used for the data analysis. Was the data key into SPSS directly or imported?

Results

Present table for the associations of parental attitudes and demographic characteristics similar to the table on knowledge section. 

Author Response

Point 1: Most upper respiratory tract infections are caused by viruses, including rhinoviruses which account for at least 30% of cases. This is an interesting figure; however, it will be important to categorically state if this substantial proportion is children or across all ages. 

Response 1: I sincerely thank you in providing us so many tailored and good comments. For this, it should be under 5 years of age.

Point 2: For a smooth and proper introduction flow, the transmission of the virus was highlighted in lines 40-42. Similarly, the transmission was further alluded to in Lines 56-58. I suggest you develop statements on the same topic in the same paragraph.   

Response 2: Thank you for raising this point.  We have moved all details of covid-19 to the same paragraph for continuity. 12/12 

Point 3: The sample size of 385 calculated in the study was based on household figures. Clarify if the 385 sample size refers to households or the number of parents or even children.  

Response 3: It should be parents.

Point 4: Section IV of the survey instrument in Lines 151 to 156 is supposed to be section I as it is the items that the participant will answer first.   

Response 4: In Hong Kong, it is quite common that survey (government or acaedemic) might ask demographic data at the final stage (probably due to the culture that Hong Kong people are busy and surveyors sometimes may want to ask those important questions first in the survey) and our study used this survey design/flow too. 

Point 5: The knowledge section had 12 items, and 10 were true or false. It will be essential if the response options of the 2 other items in the knowledge section are stated or based on what criteria they are answered.   For the attitude and practice sections, the score for each correct response needs to be stated.  

Response 5: Please accept our apologies that, since we did not have the detailed background of each participants, it could be difficult to say the highest frequency of hand hygiene or household cleaning is directly equal to correct response, unlike knowledge which may be easily stated as correct or not.  So our study can only say parents practice more or less or got a higher score or lower score, but not meant parents did higher score is equal to they do the correct things (note: sometimes do too much/frequent hygiene practices which serve more than needed and cause burden to usual daily activity might even be harmful to psychological health). 

Point 6: State the total scores obtainable (minimum and maximum) for each of the knowledge, attitude and practice sections.  

Response 6: Thanks for your comments, these scores are stated in the revised manuscript.

Point 7: Similarly, for all the sections, how many questions are constructed as positive statements and how many are negative statements (reverse statements)? 

Response 7: Thanks for your comments, these scores are stated in the revised manuscript.

Point 8: IBM SPSS Statistics for Windows, version 26 was used for the data analysis. Was the data key into SPSS directly or imported? 

Response 8: Data key into SPSS imported.  

Point 9: Present table for the associations of parental attitudes and demographic characteristics similar to the table on knowledge section.  

Response 9: We apologies that we would like to keep the current content as we would like to let the manuscript to focus on the knowledge gaps rather than associations.